# Illumina complete long read assay yields contiguous bacterial genomes from human gut metagenomes

Dylan G. Maghini,[1] Yuya Kiguchi,[1] Aaron E. Darling,[2,3] Leigh G. Monahan,[2] Aaron L. Halpern,[2] Catherine M. Burke,[4] Erich Jaeger,[2] Aaron Statham,[2] Tiffany Truong,[2] Kevin Ying,[2] Stephen P. Bruinsma,[2] Gary P. Schroth,[2] Ami S. Bhatt[5,6]

**ABSTRACT** Metagenomics enables direct investigation of the gene content and potential functions of gut bacteria without isolation and culture. However, metagenome-assembled genomes are often incomplete and have low contiguity due to challenges in assembling repeated genomic elements. Long-read sequencing approaches have successfully yielded circular bacterial genomes directly from metagenomes, but these approaches require high DNA input and can have high error rates. Illumina has recently launched the Illumina Complete Long Read (ICLR) assay, a new approach for generating kilobase-scale reads with low DNA input requirements and high accuracy. Here, we evaluate the performance of ICLR sequencing for gut metagenomics for the first time. We sequenced a microbial mock community and 10 human gut microbiome samples with standard, shotgun $2 \times 150$ paired-end sequencing, ICLR sequencing, and nanopore long-read sequencing and compared performance in read lengths, assembly contiguity, and bin quality. We find that ICLR human metagenomic assemblies have higher N50 ($119.5 \pm 24.8$ kilobases) than short read assemblies ($9.9 \pm 4.5$ kilobases; $P = 0.002$), and comparable N50 to nanopore assemblies ($91.0 \pm 43.8$ kilobases; $P = 0.32$). Additionally, we find that ICLR draft microbial genomes are more complete ($94.0\% \pm 20.6\%$) than nanopore draft genomes ($85.9\% \pm 23.0\%$; $P \leq 0.001$), and that nanopore draft genomes have truncated gene lengths ($924.6 \pm 114.7$ base pairs) relative to ICLR genomes ($954.6 \pm 71.5$ base pairs; $P \leq 0.001$). Overall, we find that ICLR sequencing is a promising method for the accurate assembly of microbial genomes from gut metagenomes.

**IMPORTANCE** Metagenomic sequencing allows scientists to directly measure the genome content and structure of microbes residing in complex microbial communities. Traditional short-read metagenomic sequencing methods often yield fragmented genomes, whereas advanced long-read sequencing methods improve genome assembly quality but often suffer from high error rates and are logistically limited due to high input requirements. A new method, the Illumina Complete Long Read (ICLR) assay, is capable of generating highly accurate kilobase-scale sequencing reads with minimal input material. To evaluate the utility of ICLR in metagenomic contexts, we applied short-read, long-read, and ICLR methods to simple and complex microbial communities. We found that ICLR outperforms short-read methods and yields comparable metagenomic assemblies to standard long-read approaches while requiring less input material. Overall, ICLR represents an additional option for assembling complete genomes from complex metagenomes.

**KEYWORDS** metagenomics, sequencing, microbiome, genomics

The ability to generate complete and accurate microbial genomes is fundamental to understanding how microbes interact within their communities and environment. Despite extensive efforts to catalog microbial genome diversity (1, 2), most bacterial

**Peer Reviewer** Congying Chen, Jiangxi Agricultural University, Nanchang, Jiangxi, China

Address correspondence to Ami S. Bhatt, asbhatt@stanford.edu.

Dylan G. Maghini and Yuya Kiguchi contributed equally to this article. Authorship order was determined based on duration of contribution to the work.

A.E.D., L.G.M., A.L.H., E.J., A.S., T.T., K.Y., S.P.B., and G.P.S. are current or former employees of Illumina, Inc.

See the funding table on p. 9.

genomes assembled from metagenomic short-read sequencing are fragmented due to the repetitive elements and strain heterogeneity present in complex microbiomes. The use of long-read sequencing approaches, such as Oxford Nanopore and PacBio, has successfully yielded circular bacterial genomes from gut metagenomes (3–6). However, very few large-scale long-read studies have been conducted to date, as long-read approaches are still limited by their high input mass requirements and high but improving error rates (7).

Synthetic long-read (SLR) approaches, which involve the computational generation of long reads via library preparation methods that barcode reads generated from the same parent molecule, provide an alternative to true long-read sequencing. SLR and read cloud methods such as TruSeq SLR, LoopSeq, and 10× Genomics linked-reads have successfully generated full-length bacterial 16S amplicons and contiguous bacterial genome assemblies (8–10) and enabled strain-level tracking in microbial communities (11). While SLR and read cloud approaches typically have higher per-base accuracy and lower input mass requirements than long-read methods (9, 12–14), many SLR methods require intensive partitioning through dilution into well plates or separation into droplets, which limits their ability to effectively assemble bacterial taxa and common repetitive elements that are too abundant for effective dilution.

Illumina has recently announced a new method, the Illumina Complete Long Read (ICLR) assay, for long-range genome sequencing. Unlike earlier linked read and SLR methods, ICLR does not require physical partitioning of the sample DNA, enabling a simpler and streamlined sample processing workflow. Instead, the ICLR workflow uses nucleotide analogs to randomly "mark" long fragments during an early PCR step. Individual marked long fragments are amplified, fragmented, and sequenced. The resulting marked short reads are then used to informatically reconstruct the original full contiguous long fragment. This approach permits integration of long read generation within a relatively standard short read sample preparation workflow (Fig. S1) (15, 16). The ICLR method has been optimized for human genomic applications; however, sub-assembled ICLR reads are promising for metagenomic assembly, as the reported read N50s of 6–7 kilobases are long enough to span insertion sequences, ribosomal RNA genes, and other common repetitive bacterial genomic elements. Here, we apply the ICLR assay to a microbial mock community and human gut metagenomes to quantify the assay's ability to generate contiguous microbial genomes from metagenomes.

## RESULTS AND DISCUSSION

We first sought to evaluate the ability of ICLR technology to assemble complete microbial genomes from a microbial mock community, relative to standardized short- and long-read sequencing methods. We performed ICLR and Oxford Nanopore Technologies (ONT) long-read sequencing on the ZymoBiomics HMW DNA Standard D6322, which is a mixture of high-molecular-weight DNA from seven bacterial and one yeast species. We obtained 397 Gbp of ICLR short reads using an early version of the ICLR chemistry (see Materials and Methods), which were then assembled into 49.6 gigabase pairs (Gbp) of long sub-assembled reads (Fig. S2) with an N50 of 7.5 kbp. We obtained 10.9 Gbp of ONT long reads with a read N50 of 4.5 kbp after basecalling and used previously published (7) paired-end short reads (7.5 Gbp) from the mock community (Table S1). The ICLR workflow includes a dilution step that limits the number of genome equivalents carried through (maximum coverage depth) and the amount of sequencing required. This can be simulated *in silico* by downsampling ICLRs from the full data sets generated here. To compare the performance of these sequencing approaches, we randomly subsampled each sequencing data set to 0.5, 1, 2, 5, and 10 Gbp (ONT long reads and ICLR sub-assembled reads) or 0.5, 1, 2, 5, and 7 Gbp (short reads) ten times prior to assembly and manual binning of contigs into draft genomes (see Materials and Methods). Notably, the ICLR method first required sub-assembly of short reads into long reads, thus each long read represents a much higher depth of short read coverage. However, as the total depth of short reads needed to assemble microbial long fragments

has not been optimized in the ICLR assay, the following analyses consider sequencing depth measured by sub-assembled reads only.

Assembly of the mock community illustrates that the ICLR sub-assembled reads are able to assemble into accurate, highly contiguous microbial genomes from simple metagenomes. Total assembly length increases with sequencing depth for each sequencing method, with ICLR sub-assembled reads reaching the expected assembly length of 41.1 megabase pairs (Mbp) with 5 Gbp of long reads (Fig. 1a; Data S1). Assembly contiguity plateaus at low sequencing depths for all methods (Fig. 1b). While ONT assemblies have higher contiguity than the ICLR assemblies, both methods achieve megabase-length contiguity. Unpolished ONT assemblies have higher indel rates than ICLR assemblies (Fig. 1c). High indel rates are a known issue in ONT sequencing and can be resolved with short-read polishing after assembly (Fig. 1c), but this step requires additional sequencing and computational processing. Binned contigs reveal that ONT and ICLR reads assemble microbial genomes to comparable contiguity at all read depths for most members of the mock community (Fig. 1d; Data S2). The exceptions are *Escherichia coli* and *Salmonella enterica*, for which ONT bins have higher contiguity. This disparity is likely due to the high incidence of insertion sequences in the *E. coli* and *S. enterica* genomes (Table S2). However, ICLR assemblies maintain higher contig

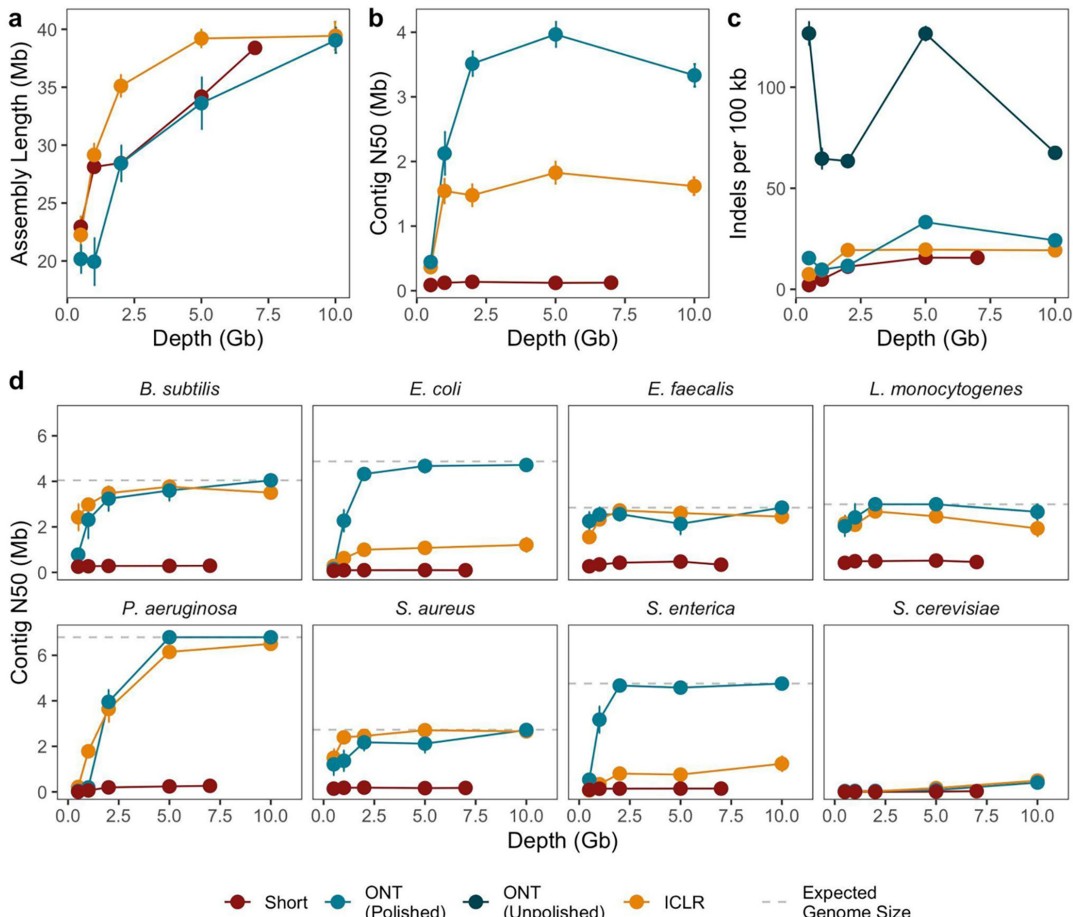

**FIG 1** Illumina Complete Long-Read (ICLR) assay performance on a microbial mock community. Sequencing reads from 2 × 150 paired-end sequencing (Short), ICLR sequencing, and Oxford Nanopore Technologies (ONT) long-read sequencing were subsampled ten times to five read depths, followed by assembly and binning of contigs into draft genomes. Assemblies were evaluated for (a) assembly length in megabases contained in contigs greater than 10,000 base pairs, (b) contig N50 in megabases, and (c) indels per 100 kilobases relative to mock community reference genomes. (d) Contig N50 in megabases for draft genomes. Points represent mean value across ten random subsamples (a–c) or all bins per organism recovered from 10 random subsamples (d). Whiskers represent standard error of the mean. Dashed lines represent expected genome size per organism (not shown for *S. cerevisiae*, which has expected genome size of 12.1 megabases). Statistics for unpolished nanopore assemblies are shown in panel c.

N50 than that of short-read assemblies for these two organisms. Finally, all methods generated *Saccharomyces cerevisiae* genomes with a low contig N50, even though the average depth of the *S. cerevisiae* was 20-fold depth in 10 Gbp subsampled ICLR. The ICLR assembly graph corresponding to *S. cerevisiae* showed many nodes connected by multiple edges, suggesting that the read length of ICLR is not long enough to resolve the genomic complexity of *S. cerevisiae* (Fig. S3). Higher counts of misassemblies in ICLR assemblies relative to other methods are largely attributable to misassembly of *S. cerevisiae* (Data S2). Overall, we observe that ICLR improves greatly on short read methods alone, while yielding contigs that are slightly less contiguous but more accurate than ONT long reads.

Next, we evaluated the performance of the ICLR method on complex microbiomes. We collected 10 human fecal samples from healthy adult volunteers in California and sequenced with each method; we obtained a mean of 14.02 (range: 6.97–32.2) Gbp of short reads, 9.03 (range: 3.82–13.7) Gbp of ONT long reads, and 18.10 (range: 15.11–20.88) Gbp of ICLR sub-assembled reads from a mean of 312 (range: 267–384) Gbp of marked ICLR sequencing and 123.75 (range: 107–137) Gbp of unmarked reads (Fig. S4; Table S3), which were generated using an early version of the ICLR chemistry (see Materials and Methods). Metagenomic assembly yields total assembly lengths that are comparable between short-read and ICLR assemblies, while the ICLR assemblies are longer than ONT long-read assemblies (Fig. 2a; adjusted *P* = 0.01; paired Wilcoxon signed-rank test corrected by Benjamini-Hochberg (BH); Data S3). The ICLR assemblies are more contiguous than short-read assemblies (Fig. 2b; adjusted *P* = 0.007; paired Wilcoxon signed-rank test corrected by BH) and have comparable contiguity to the ONT long-read assemblies (adjusted *P* = 0.28; paired Wilcoxon signed-rank test corrected by BH). However, while contig N50 is comparable between both long-read methods,

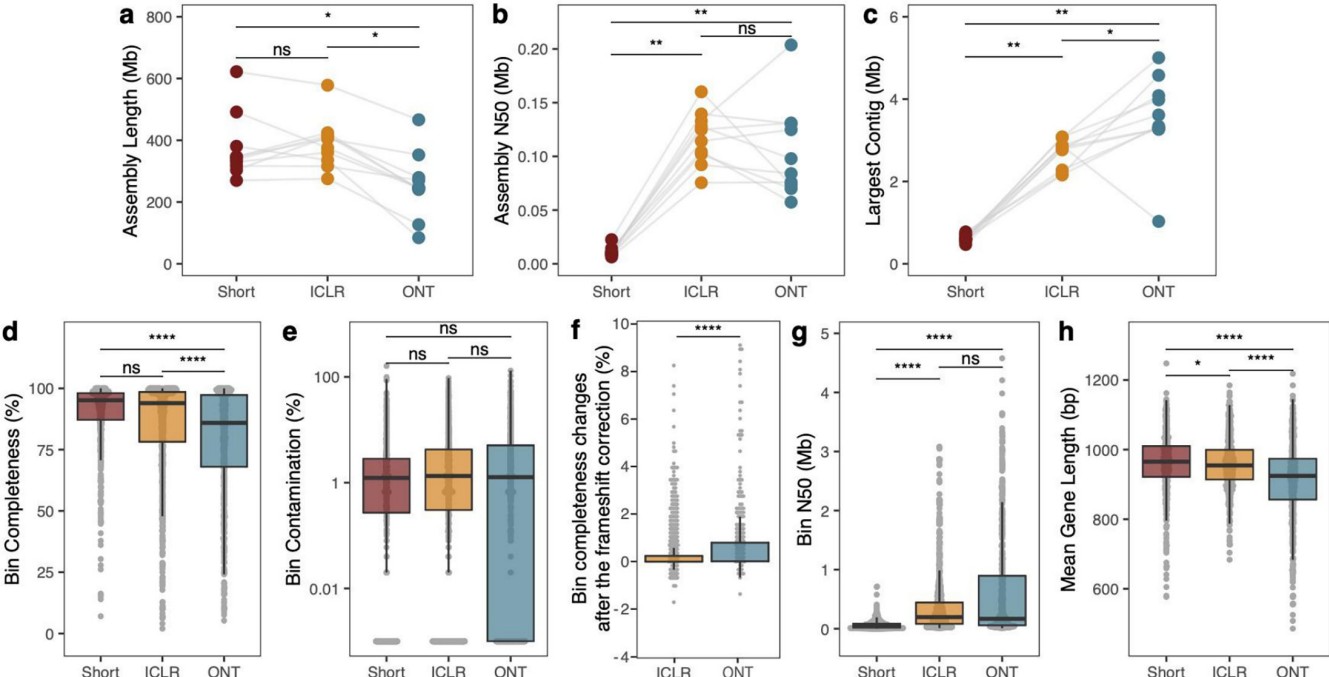

**FIG 2** Illumina Complete Long-Read (ICLR) assay performance on human gut metagenomes. Assemblies and bins from human metagenomes sequenced with 2 × 150 paired-end sequencing (Short), ICLR sequencing, and Oxford Nanopore Technologies (ONT) long-read sequencing. Assemblies were evaluated for (a) total assembly length in megabases contained in contigs greater than 500 base pairs, (b) contig N50 in megabases, and (c) largest contig contained in the assembly. Points represent values for each sample (*n* = 10). Bins were evaluated for completeness (d) and contamination (e) based on the presence of single-copy marker genes. The effect of frameshift for bin's completeness in ICLR and ONT was evaluated by comparing the changes in completeness before and after frameshift correction (f). The contiguity of bins was evaluated in N50 megabases (g) and mean gene length per bin (h). Points represent values for individual bins. Box plots indicate first, second, and third quartiles, with whiskers indicating the lowest and highest value no further than 1.5 times the interquartile range.

the largest contigs in ONT assemblies are longer than those in ICLR assemblies (Fig. 2c; adjusted $P$ = 0.047; paired Wilcoxon signed-rank test corrected by BH). After reference-agnostic binning each assembly into putative draft genomes (see Materials and Methods), we find that ICLR bins are more complete (Fig. 2d; adjusted $P \leq 0.001$; Wilcoxon rank-sum test corrected by BH; Data S4) and comparably contaminated (Fig. 2e; adjusted $P$ = 0.47; Wilcoxon rank-sum test corrected by BH) as ONT bins, measured by the presence of single-copy core genes. The improvement in bin completeness after correction of frameshift error (17) was significantly higher for ONT than for ICLR (Fig. 2f; $P \leq 0.001$; Wilcoxon rank-sum test), suggesting that the difference in bin completeness between ICLR and ONT is due to undercounting of single-copy core genes in nanopore assemblies due to frameshift errors. There is no difference in bin contiguity between the ICLR and ONT long-read methods (Fig. 2g; adjusted $P$ = 0.61; Wilcoxon rank-sum test corrected by BH). As expected, the mean gene length per bin is higher in ICLR bins than in ONT long-read bins (Fig. 2h; adjusted $P \leq 0.001$; paired Wilcoxon signed-rank test corrected by BH). Finally, the ICLR method is capable of yielding single-contig, megabase scale draft genomes (Table S3). These results indicate that in real-life sequencing applications, the ICLR assay is capable of yielding high-quality metagenomic assemblies and bins, improving greatly on the contiguity of short-read approaches and performing competitively with best-practice long-read methods.

Given the strong performance of the ICLR assay, the limitations and potential of this assay merit further evaluation. Here, we performed extremely high-depth ICLR sequencing to fully evaluate the potential of ICLR to generate contiguous metagenomic assemblies. Subsampling ICLR and Oxford Nanopore long reads from the microbial mock community allowed for a comparison of assembly outcomes based on read lengths and accuracy. We find that ICLR has the potential to overcome the technical limitations of conventional short and ONT read-based metagenomic assembly, such as low continuity and high error rate, respectively. However, in order to evaluate the general use of this method, it will be necessary to evaluate the practical marked and unmarked short-read depths in the future, which will vary depending on the research design. Further, a comparison of experimental feasibility will require official pricing of library preparation kits and sequencing costs.

Another limitation of this study is the difficulty of assembling fungal genomes from metagenomic samples in ICLR, despite relatively high sequencing coverage. We speculate that the read length of ICLR is insufficient for spanning repeats in the *S. cerevisiae* genome. The existence of Ty repeat elements in *S. cerevisiae*, approximately 6 kbp long and with high identity between copies (18), likely contributed to shorter assemblies and increased mis-assemblies. Reconstruction of fungal genomes with high completeness may require longer reads than are produced by the current ICLR protocol. Our results provide a preliminary assessment of the potential of the ICLR assay for metagenomic applications, but future research is needed to evaluate the specific genomic and technical factors that limit assembly contiguity using market-ready versions of the ICLR assay.

True long-read sequencing methods have been on the market for several years and were applied to gut microbiomes 4 years ago (3, 6), but only a small number of studies have leveraged these techniques for large-scale metagenomic studies. In part, this limited uptake may be due to the high input requirements and hands-on nature of existing long-read platforms. The ICLR assay captures long-range information on existing sequencing platforms with low input requirements. Here, we sought to evaluate the performance of ICLR and contextualize these results with field-standard short- and long-read sequencing approaches. We observe that the ICLR assay can generate highly contiguous and accurate bacterial genome assemblies from simple and complex metagenomes. We expect that the assay will be a scalable solution for metagenomics studies: the assay offers the distinct advantage of yielding standard high-accuracy paired-end short reads as well as long reads, which allows for the use of standard metagenomic analysis pipelines while enabling more contiguous assembly

and binding. Therefore, contingent on reagent pricing and sequencing costs, the ICLR assay represents a promising new technique for metagenomics. A whitelist app for ICLR marked read subassembly and rendering is now available for limited use on Illumina Connected Analytics for individuals seeking to perform custom reference analysis (see Materials and Methods).

## MATERIALS AND METHODS

### Sample collection

Ten human subjects were enrolled under Stanford IRB 42043 (PI: Ami S. Bhatt) and informed consent was obtained from all participants. Single fecal samples were collected from all participants and stored immediately in 2 mL cryovials at −80°C without a preservative buffer.

### Sequences of Illumina short reads from mock community

Short-read sequencing data for the ZymoBiomics HMW DNA Standard (Zymo Research) were downloaded from the European Nucleotide Archive from Project PRJEB48692 and used as a representative short-read data set for the mock community.

### Sequences of Nanopore long reads from mock community

Nanopore sequencing libraries on the mock community were prepared from the ZymoBiomics HMW DNA Standard using the Q20+ SQK-LSK112 Kit (ONT) according to the manufacturer's instructions and sequenced on one MinION R9.4.1 flow cell.

### Sequencing of ICLR from mock community

ICLR libraries (Illumina, Inc. for research use only) were prepared from the mock community according to the manufacturer's instructions, with the following exceptions. First, tagmented DNA was diluted to 8 pg/µL at step 7 of the Dilute Tagmented DNA Procedure. Second, PCR cycling was increased to 16 cycles at step 2 of the Amplify Diluted DNA Preparation. The ICLR library was sequenced to 2.64 B reads on a NovaSeq 6000 S2 flowcell at 2 × 150 bp read length at a final loading concentration of 200 pM. 40.4 M short reads were generated from an Illumina DNA PCR-Free library (Illumina, Inc. for research use only) prepared according to the manufacturer's instructions, and Megahit v1.2.9 was used for *de novo* assembly. These short read assemblies were then used to inform ICLR *de novo* assemblies done with Flye v2.9-b1768.

### DNA extraction from human fecal samples

DNA was extracted from 250 mg of each stool sample using the QIAamp PowerFecal Pro DNA Kit (Qiagen) according to the manufacturer's instructions, with the exception of using the EZ-Vac Vacuum Manifold (Zymo Research) in place of centrifugation for all steps except for the final elution. DNA concentration was measured using a Qubit 3.0 fluorometer (Thermo Fisher Scientific) with the dsDNA High Sensitivity Kit.

### Sequencing of short reads from human fecal metagenomic DNA

Short-read sequencing libraries on human stool samples were prepared using the Illumina DNA Prep Kit (Illumina, Inc. for research use only). Libraries were pooled in equal concentration and sequenced on a NovaSeq 6000 (Illumina, Inc. for research use only) at 2 × 150 reads.

### Sequencing of Nanopore long reads from human fecal metagenomic DNA

Nanopore sequencing libraries on human stool samples were prepared using the Q20+ SQK-NBD112.24 Kit (ONT) according to the manufacturer's instructions. Libraries were

pooled in two sets of five samples, and each sample pool was sequenced on two PromethION R9.4.1 flow cells (ONT). Nanopore reads were basecalled and demultiplexed using Guppy v5.1.13. The same sample-derived reads from the two flow cells were pooled and used for subsequent analysis.

## Sequencing of ICLR from human fecal metagenomic DNA

ICLR libraries were prepared using an initial version of the protocol that was subsequently modified extensively to create the final commercialized product. As such, there were a number of changes from the current published protocol, which are summarized at the end of this section. To tagment genomic DNA, a solution-based rather than bead-based system was used that required different reagents and reaction volumes, and a 0.6× SPRIselect step was used for the Post-Tagmentation Clean Up. To mark tagmented DNA, LMM, LPM, and LRP1 were used in different ratios to accommodate 10 μL of the tagmented DNA in solution rather than on-bead, and five cycles of PCR were used for initial amplification. SPRIselect beads were used to clean up and elute the amplified product into 25 μL, which was then amplified in a second similar PCR that omitted LMM and the initial 68°C incubation. SPRIselect beads were again used to clean up and elute the amplified product into 25 μL. The tagmented DNA was then diluted to 21.6 pg/μL and amplified with LPM and LRP2. PCR was done for 16 cycles, and SPRIselect beads were used to clean up and elute the amplified product into 25 ul. To fragment long templates, 100 ng of DNA product was tagmented with BLT's, and the TAG program 55°C incubation was extended from 5 to 15 min. To append index adapters, TSB from the Illumina DNA Prep Kit was used in place of ST2, and the reaction was incubated at 37°C for 15 min. TWB from the Illumina DNA Prep Kit was also used in place of TWB2, and an earlier developmental version of LPM was used in place of EPM4. In addition, the INDEX PCR program was changed to the following: 68°C for 3 min, 6 cycles of (98°C at 10 s, 55°C at 15 s, and 68°C at 1 min), 68°C at 1 min, and a 4°C hold. For the final library clean up, a 0.6× SPRIselect clean up was followed with a 0.4–0.15× double-sided size selection. All libraries were sequenced on a NovaSeq 6000 S4 flowcell at 2 × 150 bp read length with a final loading concentration of 0.75 pM.

Although the fundamental library prep chemistry is the same, some differences in the above steps were implemented in the on-market kit as part of the standard product development optimization process. In contrast to the current work, for the on-market kit: (i) initial tagmentation is performed on bead (BLT-LR), then after two washes, marking PCR occurs in a single step (omitting the initial five cycle PCR listed above) directly off the BLT-LR to improve simplicity, robustness, and yield; (ii) Illumina Purification Beads are used for size selection and cleanup instead of SPRIselect beads to align with similar Illumina workflows; (iii) volumes are adjusted to enable more robust performance (e.g., elutions in 100 μL instead of 25 μL); (iv) instead of 100 ng DNA input into the "fragment long templates" step, a set volume (30 μL) of DNA is used, taking advantage of BLT saturation by small amounts of DNA, improving robustness; (v) TWB2 is used in place of TWB to align with similar Illumina workflows; (vi) EPM4 is used in place of LPM for the final PCR amplification, and optimized thermocycling parameters are used to optimize and align with similar Illumina workflows; (vii) ST2 is used in place of TSB, and incubation is for 2 min at room temperature to simplify, improve robustness, and align with similar Illumina workflows; and (viii) the final SPRI size selection cleanup is changed to a two-sided 0.4×/0.6× cleanup to simplify and optimize for robustness.

## Sub-assembly of ICLR

Marked and unmarked reads were processed using an early-stage development version of the ICLR data analysis software. Unmarked reads were first assembled using MEGAHIT v1.2.9. The resulting draft assembly was used as a reference for ICLR construction. Briefly, ICLR construction consists of mapping marked reads to the assembly, identifying the marked positions in each read by comparison to the assembly, grouping the reads which share common marker patterns, assembling the grouped reads, and finally removing the

markers by comparison to the unmarked reads. A description of the detailed software analysis workflow is available online (15).

## Metagenomic assembly and binning

Mock community reads were subsampled to 0.5, 1, 2, 5, and 7 Gb (short reads) and 10 Gb (ICLR and ONT reads) 10 times using seqtk v1.3-r106 (19). Short reads were assembled using SPAdes v3.15.3 using the --meta flag. ICLR and ONT reads were assembled using Lathe v1 (3) as described previously (20). Briefly, Lathe performs assembly with metaFlye 2.4.2 (21) and optional short-read polishing using publicly available short reads (European Nucleotide Archive from Project PRJEB48692) with Pilon v1.23 (22). Lathe also performed misassembly detection and correction by breaking locations in the assembly that are covered by zero or one long read. For all methods, contigs were binned into draft genomes by aligning to the reference genomes for each organism, which are available from Zymo Research. Alignments were performed with minimap2 v2.24 (23), and primary alignments were selected using samtools v1.16.1 (24).

Human gut metagenome short reads and nanopore long reads were assembled as described above, without subsampling. ICLR sub-assembled reads were assembled with metaFlye 2.4.2. Contigs were binned using MetaBAT2 v2.15 (25), CONCOCT v1.1.0 (26), and MaxBin 2.0 v2.2.7 (27), and bins were aggregated for each sample using DAS Tool v1.1.2 (28). The frameshift error of Bins was corrected by proovframe (17) with the prokaryotic marker gene database of CheckM2 (29).

Assembly quality was evaluated with QUAST 5.2.0, with comparison to reference genomes included for mock community evaluation. Bin completeness and contamination were measured using CheckM v1.0.13 (30). Bins that had >90% completeness and <5% contamination were classified as high-quality bins, and bins that had ≥50% completeness and <10% contamination were classified as medium quality bins, based on the Genome Standards Consortium (31). Gene counts and lengths were measured with prokka v1.14.6 (32). Insertion sequences were detected with ISEscan v1.7.2.3 (33). The assembly graph was visualized using Bandage (34). To identify the graph corresponding to *S. cerevisiae*, the contig sequences in the assembly graph were aligned to the *S. cerevisiae* reference genome using minimap2 v2.24 (23).

## Statistical analysis and plotting

Plotting was performed in R v4.1.2 with packages ggplot2 v3.3.5 (35), tidyverse v1.3.1 (36), reshape2 v1.4.4 (37), cowplot v1.1.1 (38), and paletteer v1.4.0 (39). For plotting bin contamination, a pseudocount of 0.001 was added to bins with no contamination to allow plotting on a log scale.

## ACKNOWLEDGMENTS

We thank the study participants for volunteering to participate in this study. We thank members of the Bhatt lab for technical assistance. ICLR kits and analysis support for the ICLR read construction were supplied by Illumina, Inc.

This work was supported in part by NIH grant P30 CA124435, which supports the Stanford Cancer Institute Genetics Bioinformatics Service Center. Computing costs were also supported, in part, by aan NIH S10 Shared Instrumentation Grant 1S10OD02014101. This work was supported in part by NIH R01AI148623 and R01AI143757, a Stand Up 2 Cancer Grant, the Chan Zuckerberg Initiative, a Sloan Foundation Fellowship, and the Allen Distinguished Investigator Award (to A.S.B.). D.M. was supported by the Stanford Graduate Fellowships in Science and Engineering Program and the Stanford Gerald. J. Lieberman Fellowship. Y.K. is supported by a Stanford MCHRI Pediatric IBD post-doctoral fellowship.

A.S.B., G.P.S., and D.G.M.. conceptualized the study. D.G.M. designed the study, collected samples, and performed DNA extraction on all samples. T.T. and E.J. performed ICLR library preparation and sequencing, and D.G.M. performed nanopore library

preparation and sequencing. A.E.D., L.G.M., A.L.H., S.P.B., and C.M.B. contributed to the conceptualization and design of ICLR methods for metagenomic applications and analysis of pilot ICLR metagenomic data. A.L.H., A.S., and K.Y. executed ICLR data analysis. A.L.H. performed ICLR application design and data wrangling. D.G.M. and Y.K. carried out analyses and generated figures. D.G.M., Y.K., and A.S.B. wrote the manuscript. All authors read and approved the final manuscript.

## AUTHOR AFFILIATIONS

[1]Department of Medicine (Hematology), Stanford University, Stanford, California, USA
[2]Illumina Inc, San Diego, California, USA
[3]Australian Institute for Microbiology and Infection, University of Technology Sydney, Ultimo, New South Wales, Australia
[4]University of Technology Sydney, Ultimo, New South Wales, Australia
[5]Department of Medicine (Hematology, Blood and Marrow Transplantation), Stanford University, Stanford, California, USA
[6]Department of Genetics, Stanford University, Stanford, California, USA

## AUTHOR ORCIDs

Dylan G. Maghini  http://orcid.org/0000-0001-9542-492X
Yuya Kiguchi  http://orcid.org/0000-0003-3334-2018
Ami S. Bhatt  http://orcid.org/0000-0001-8099-2975

## FUNDING

| Funder | Grant(s) | Author(s) |
| --- | --- | --- |
| National Institutes of Health | P30 CA124435, R01 AI143757, 1S10OD02014101, R01 AI148623 | Ami S. Bhatt |
| Alfred P. Sloan Foundation | | Ami S Bhatt |
| Stand Up To Cancer | | Ami S. Bhatt |
| Chan Zuckerberg Initiative | | Ami S. Bhatt |
| Stanford Medicine Children's Health Center for IBD and Celiac Disease | | Yuya Kiguchi |
| Allen Foundation | | Ami S. Bhatt |

## AUTHOR CONTRIBUTIONS

Dylan G. Maghini, Conceptualization, Data curation, Formal analysis, Investigation, Methodology, Project administration, Resources, Visualization, Writing – original draft, Writing – review and editing | Yuya Kiguchi, Formal analysis, Investigation, Methodology, Visualization, Writing – review and editing | Aaron E. Darling, Conceptualization, Methodology, Software, Supervision, Writing – review and editing | Leigh G. Monahan, Conceptualization, Investigation, Methodology, Writing – review and editing | Aaron L. Halpern, Investigation, Methodology, Software, Writing – review and editing | Catherine M. Burke, Investigation, Software, Writing – review and editing | Erich Jaeger, Investigation, Writing – review and editing | Aaron Statham, Investigation, Writing – review and editing | Tiffany Truong, Investigation, Writing – review and editing | Kevin Ying, Investigation, Writing – review and editing | Stephen P. Bruinsma, Investigation, Project administration, Resources, Supervision, Writing – review and editing | Gary P. Schroth, Conceptualization, Funding acquisition, Investigation, Project administration, Resources, Supervision, Writing – review and editing | Ami S. Bhatt, Conceptualization, Funding acquisition, Supervision, Writing – original draft, Writing – review and editing

## DATA AVAILABILITY

All sequencing data generated are available on the NCBI Sequence Read Archive under BioProject PRJNA940499. Short-read sequencing data for the ZymoBiomics HMW DNA Standard was previously published (7) and are available on the European Nucleotide Archive at project PRJEB48692. Workflows for metagenomic assembly and binning are available at https://github.com/bhattlab/bhattlab_workflows and https://github.com/bhattlab/lathe. Analysis and plotting scripts can be found at https://github.com/dgmaghini/ICLRMetagenomics/. Standard, supported ICLR applications are focused towards human genome applications, where marked reads are mapped to a human genome reference and subsequently grouped by common marker patterns and assembled, then combined with unmarked reads to remove marker patterns. A whitelist application is now available via Illumina Connected Analytics for restricted use by users interested in performing ICLR assembly with a custom reference genome. This application takes unmarked and marked FASTQ files and a user-provided reference genome FASTA as input and produces a FASTQ of ICLRs for downstream use. For metagenomic analyses similar to those conducted here, users can provide a *de novo* metagenomic assembly of unmarked reads along with the original marked and unmarked reads to generate ICLRs. This application currently is unsupported and under development, and source code is not publicly available. To request access contact Colin Davidson (cdavidson@illumina.com) or Illumina technical support (techsupport@illumina.com).

## ADDITIONAL FILES

The following material is available online.

### Supplemental Material

**Figure S1 (mSystems01531-24-s0001.pdf).** ICLR library preparation workflow.
**Figure S2 (mSystems01531-24-s0002.pdf).** Read lengths from microbial mock community sequencing.
**Figure S3 (mSystems01531-24-s0003.pdf).** Metagenomic assembly graph corresponding to *S. cerevisiae* genome.
**Figure S4 (mSystems01531-24-s0004.pdf).** Read lengths from human gut metagenomic sequencing.
**Supplemental Data (mSystems01531-24-s0005.xlsx).** Data S1 to S4.
**Supplemental Tables (mSystems01531-24-s0006.xlsx).** Tables S1 to S3.

### Open Peer Review

**PEER REVIEW HISTORY (review-history.pdf).** An accounting of the reviewer comments and feedback.

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
