## [Reviewer comments · mSystems]

Illumina Complete Long Read assay yields contiguous bacterial genomes from human gut metagenomes

Dylan Maghini, Yuya Kiguchi, Aaron Darling, Leigh Monahan, Aaron Halpern, Catherine Burke, Erich Jaeger, Aaron Statham, Tiffany Truong, Kevin Ying, Stephen Bruinsma, Gary Schroth, and Ami Bhatt

Corresponding Author(s): Ami Bhatt, Stanford University

Review Timeline:

Submission Date:	November 14, 2024
Editorial Decision:	December 29, 2024
Revision Received:	May 10, 2025
Accepted:	June 11, 2025

Editor: Daniel Garrido

Reviewer(s): Disclosure of reviewer identity is with reference to reviewer comments included in decision letter(s). The following individuals involved in review of your submission have agreed to reveal their identity: Congying Chen (Reviewer #2)

Transaction Report:

DOI: <https://doi.org/10.1128/msystems.01531-24>

Re: mSystems01531-24 (Illumina Complete Long Read Assay yields contiguous bacterial genomes from human gut metagenomes)

Dear Prof. Ami S Bhatt:

Thank you for the privilege of reviewing your work. Below you will find my comments, instructions from the mSystems editorial office, and the reviewer comments. Both reviewers suggested minor modifications to your manuscript.

Revision Guidelines

Sincerely,
Daniel Garrido
Editor
mSystems

Reviewer #1 (Comments for the Author):

This paper describes the application of a new sequencing technology, the Illumina Complete Long Read (ICLR) assay, for assembly of microbial metagenomes. The authors claim that ICLR is more accurate than standard long-read technologies and produces more contiguous assemblies than short-read sequencing alone, and it is less labor-intensive, requires less DNA input, and is more scalable than current long-read methods. They compare the performance of standard short- and long-read metagenomic sequencing platforms to that of ICLR for a defined mixture of microbial genomic DNA and ten human gut samples. Read lengths, assembly contiguity, and bin quality are compared, along with assembly quality metrics, including completeness

and contamination. ICLR gives higher N50 values than SRS and the same N50 as ONT; ICLR also gives more complete draft genomes and less truncated gene lengths compared to ONT.

The evidence for the utility of ICLR is supported by the data, and claims of its usefulness are not oversold, as it is shown to be competitive with other long read methods. These claims would be stronger if supported by more data showing that its accuracy is maintained at feasible sequencing depths that would enable multiplexing of many samples. More explanation of the technology itself and how the workflow compares to other methods would also appeal to readers interested in trying it for their microbial metagenomic assembly analyses.

Major comments:

- L103-107: the authors state that the depth of short reads needed to assemble long reads from microbial genomes has not been optimized. However, it is unclear why they did not perform subsampling of their short reads before subassembly to see whether they could achieve similar results with a lower sequencing depth. Sequencing one sample on a full lane is infeasible for researchers, and their defined community is ideal for testing this variable. Claims of ICLR's potential for high throughput are not well supported without data showing that high quality assemblies can be achieved at lower sequencing depths (before subassembly).
- L132: more description of the source of these human gut samples is needed to contextualize the biomass and diversity of these samples. Is this from healthy adult volunteers?
- Further explanation of how the ICLR works in the introduction would be useful - what are the overlaps with Illumina short read DNA extraction, library prep, and sequencing platforms?
- If this is the first paper published on ICLR applied to microbial metagenomes (it appears to be), this would be good to state.
- L126-128: Why does ICLR perform poorly on *S. cerevisiae* if it's designed for use with the human genome, which is much larger? 2% *S. cerevisiae* is not a trivial abundance, and these samples were sequenced very deeply. The misassembly counts referenced here should also be quantified to support this claim.
- Could the authors provide more information on the differences between the early ICLR kit chemistry and what is commercially available? Is this the reason for so many modifications to the ICLR protocol?
- L150: Can the authors show that the difference in bin completeness is indeed due to undercounting of single copy genes in ONT assemblies due to frameshift errors? It is unclear why this would be expected if these are polished assemblies.

Minor critiques:

- Figure 1: Which short reads were used to polish the ONT assemblies? The previously published ones or those generated by the ICLR?
- P-value corrections should be described if carried out in Figure 2.
- Clarity of wording - what do we mean when we say LRS is low throughput? How many samples per run can you get compared to SRS? An approximate cost breakdown here between methods would help if possible.
- Add citations in the introduction for the description of how ICLR works if available
- L116: clarify that only unpolished ONT assemblies have higher indel rates than ICLR
- L220-222: It is unclear whether the data from the two flow cells used for each sample pool of 5 samples were treated as replicates or pooled together.
- Table S3: how are medium- and high-quality bins defined?
- Table S3: title should say human fecal/gut metagenome (as opposed to human metagenome)

Reviewer #2 (Comments for the Author):

In this manuscript, the authors compared the performance of the standard shotgun 2x150 paired-end sequencing, ICLR sequencing, and nanopore long-read sequencing with a microbial mock community and ten human gut microbiome samples. They suggested that the ICLR sequencing is a promising method for high-throughput and accurate assembly of microbial genomes from gut metagenomes. The study is very interesting. If ICLR sequencing continues to be optimized and commercialized, it will promote the investigation of metagenomic analysis. I have the following comments to the manuscript.

1. It should be better if the comparison of performance among ICLR assay, standard shotgun, and nanopore long-read sequencing with a microbial mock community were performed based on the sequencing depth of short reads, instead of sub-assembled reads for ICLR assay.
2. For the study in ten human gut metagenomes, a mean of 312 Gbp of marked and 123.75 unmarked reads were generated, which was the extremely higher sequencing depth than that in the standard shotgun sequencing. I doubted whether the bin N50 and the largest contigs in the standard shotgun sequencing should be improved if the same sequencing depth was used.
3. The section of "library preparation and sequencing" should be divided into three parts following ICLR sequencing, shotgun sequencing, and nanopore long-read sequencing.

Reviewer #1 (Comments for the Author):

This paper describes the application of a new sequencing technology, the Illumina Complete Long Read (ICLR) assay, for assembly of microbial metagenomes. The authors claim that ICLR is more accurate than standard long-read technologies and produces more contiguous assemblies than short-read sequencing alone, and it is less labor-intensive, requires less DNA input, and is more scalable than current long-read methods. They compare the performance of standard short- and long-read metagenomic sequencing platforms to that of ICLR for a defined mixture of microbial genomic DNA and ten human gut samples. Read lengths, assembly contiguity, and bin quality are compared, along with assembly quality metrics, including completeness and contamination. ICLR gives higher N50 values than SRS and the same N50 as ONT; ICLR also gives more complete draft genomes and less truncated gene lengths compared to ONT.

The evidence for the utility of ICLR is supported by the data, and claims of its usefulness are not oversold, as it is shown to be competitive with other long read methods. These claims would be stronger if supported by more data showing that its accuracy is maintained at feasible sequencing depths that would enable multiplexing of many samples. More explanation of the technology itself and how the workflow compares to other methods would also appeal to readers interested in trying it for their microbial metagenomic assembly analyses.

Major comments:

- L103-107: the authors state that the depth of short reads needed to assemble long reads from microbial genomes has not been optimized. However, it is unclear why they did not perform subsampling of their short reads before subassembly to see whether they could achieve similar results with a lower sequencing depth. Sequencing one sample on a full lane is infeasible for researchers, and their defined community is ideal for testing this variable. Claims of ICLR's potential for high throughput are not well supported without data showing that high quality assemblies can be achieved at lower sequencing depths (before subassembly).

We thank you for your insightful suggestion to evaluate the assemble efficacy using subsampled short reads for sub-assembly. This study aims to demonstrate the potential of ICLR to overcome the technical gaps in metagenome assembly using short and ONT reads. Therefore, we think that it is outside of this study's scope to evaluate the practicality of sequencing costs by evaluating the minimum limit of short reads for sub-assembly of ICLR. To avoid confusion about the aim of this study, we revised the writing to clearly state our paper's claims for evaluating the contiguity and accuracy of sequences of metagenomic assembly of ICLR compared with conventional methods and to avoid claims of higher throughput of ICLR than long reads. For this reason, we have removed the description that mentions the high throughput of ICLR (Line: 23, 36, 59, 187-188). Additionally, as the reviewers point out, the sequencing depth is crucial for practical use for further study. Therefore, in the revised manuscript, we stated the

necessity of further studies for evaluating the practical sequencing depth and cost for sequencing in the discussion section (Line: 168-172).

- L132: more description of the source of these human gut samples is needed to contextualize the biomass and diversity of these samples. Is this from healthy adult volunteers?

Thank you for your suggestion. Based on the reviewer's suggestion, we clarify the background of the collected samples as healthy adult volunteers in California (Line: 133-134).

- Further explanation of how the ICLR works in the introduction would be useful - what are the overlaps with Illumina short read DNA extraction, library prep, and sequencing platforms?

Thank you for your suggestion for improving our writing. We added additional sentences to explain the basic strategy to obtain the long reads using ICLR methodology (Line: 76-81).

- If this is the first paper published on ICLR applied to microbial metagenomes (it appears to be), this would be good to state.

Thank you for your suggestion for improving our writing. This is the first example of using ICLR for microbial metagenomics, so we added a clear statement of that in the introduction (Line: 26).

- L126-128: Why does ICLR perform poorly on *S. cerevisiae* if it's designed for use with the human genome, which is much larger? 2% *S. cerevisiae* is not a trivial abundance, and these samples were sequenced very deeply. The misassembly counts referenced here should also be quantified to support this claim.

Thank you for your invaluable comments related to the assembly of *S. cerevisiae*. As the reviewer pointed out, the average depth of the *S. cerevisiae* was 20-fold depth using 10Gb subsampled ICLR, which is sufficient to assemble the genome. To figure out the reason for the low completeness of the *S. cerevisiae* genome in metagenomic assembly, we visualized the assembly results of the ICLR as an assembly graph. As a result, the graph corresponding to *S. cerevisiae*, presented below, shows a lot of nodes linked with multiple edges (new Supplementary Figure 3). This result suggests that the read length of ICLR is not long enough to resolve the genomic complexity of *S. cerevisiae*, which is relatively higher than the bacteria studied here. Based on these results, we added a sentence to the results section discussing the low completeness of the *S. cerevisiae* genome (Line: 124-128) and now discuss the technical gaps of metagenomic ICLR assembly of fungi genomes (Line: 175-181), especially those known to have highly repetitive elements, such as the Ty element in *S. cerevisiae*. In addition, we added a description of the additional analysis in the method section (Line: 322-324).

S.cerevisiae

New Supplementary Figure 3: Metagenomic assembly graph corresponding to *S. cerevisiae* genome.

The assembly graph of a 10Gb subsampled ICLR was visualized by Bandage.

- Could the authors provide more information on the differences between the early ICLR kit chemistry and what is commercially available? Is this the reason for so many modifications to the ICLR protocol?

Thank you for the comment about improving our writing. We added a detailed explanation of the difference between our experimental process and market kit in the method section (Line: 272-287).

- L150: Can the authors show that the difference in bin completeness is indeed due to undercounting of single copy genes in ONT assemblies due to frameshift errors? It is unclear why this would be expected if these are polished assemblies.

Thank you for your comments related to the differences of the Bin completeness between ICLR and ONT. Previous studies have shown that it is challenging to accurately map short reads to low-accuracy contigs assembled using ONT reads. As a result of this, the short-read mapping becomes sparse or uneven and insufficiently polishes the contig (Tamburini et al., 2022, *Nature Communications*). Our additional analysis for correcting frameshifts, which does not depend on mapping-based polishing, showed that the improvement of the completeness of Bins after frameshift correction is significantly higher in the ONT Bins compared with ICLR Bins (new Figure 2F). These results indicate that the higher frameshift rate in the ONT contigs than in ICLR contigs is one of the reasons for the lower completeness of ONT Bins and also shows that ICLR can construct Bins with high sequence accuracy. Based on these results, we revised the result and method section (Line: 152-155 and 313-314).

Reference:

Tamburini, Fiona B., Dylan Maghini, Ovokeraye H. Oduaran, Ryan Brewster, Michaela R. Hulley, Venesa Sahibdeen, Shane A. Norris, et al. 2022. "Short- and Long-Read

Metagenomics of Urban and Rural South African Gut Microbiomes Reveal a Transitional Composition and Undescribed Taxa." *Nature Communications* 13 (1): 1–18.

Revised Figure 2: Illumina Complete Long-Read Assay performance on human gut metagenomes.

Assemblies and bins from human metagenomes sequenced with 2 x 150 paired end sequencing (“Short”), Illumina Complete Long-Read sequencing (“ICLR”) and Oxford Nanopore Technologies long-read sequencing (“ONT”). Assemblies were evaluated for (a) total assembly length in megabases contained in contigs greater than 500 base pairs, (b) contig N50 in megabases, and (c) largest contig contained in the assembly. Points represent values for each sample (n = 10). Bins were evaluated for completeness (d) and contamination (e) based on presence of single copy marker genes. The effect of frameshift for bin’s completeness in ICLR and ONT was evaluated by comparing the changes of completeness before and after frameshift correction (f). The contiguity of Bins were evaluated in N50 megabases (g) and mean gene length per bin (h). Points represent values for individual bins. Box plots indicate first, second, and third quartiles, with whiskers indicating the lowest and highest value no further than 1.5 times the inter-quartile range.

Minor critiques:

- Figure 1: Which short reads were used to polish the ONT assemblies? The previously published ones or those generated by the ICLR?

Thank you for your suggestion for improving our writing. We used previously published short reads, so we revised the method section to clarify this methodology (Line: 303-304).

- P-value corrections should be described if carried out in Figure 2.

Thank you for your suggestion related to our statistical analysis. We adjusted the p -value using the Benjamini-Hochberg method for Figure 2 and edited the Figures. Although the conclusions did not change extensively as a result of the reanalysis, the writing was revised to reflect the adjusted p -value (Line: 141-159).

- Clarity of wording - what do we mean when we say LRS is low throughput? How many samples per run can you get compared to SRS? An approximate cost breakdown here between methods would help if possible.

Thank you for your important comment. As you pointed out, the definition of low throughput varies depending on the research design, and the throughput of long read sequencing has improved significantly in recent years. Therefore, we particularly focused on our paper's claims to evaluate the contiguity and accuracy of sequences of metagenomic assembly of ICLR compared with short and ONT reads and removed the description that mentions the high throughput of ICLR (Line: 23, 36, 59, 187-188). We have avoided mentioning the cost of each method in this study because it is difficult to estimate accurate costs of every sequencing method, as these are changing quite rapidly.

- Add citations in the introduction for the description of how ICLR works if available

Thank you for the comments related to references. We added the preprint paper and technical explanation web page of ICLR to the reference list (Reference: 15 and 16).

- L116: clarify that only unpolished ONT assemblies have higher indel rates than ICLR

Thank you for the comment about improving our writing. In the revised manuscript, we now state that higher indel rates are only observed in unpolished ONT assemblies. (Line: 115).

- L220-222: It is unclear whether the data from the two flow cells used for each sample pool of 5 samples were treated as replicates or pooled together.

Thank you for the comment related to the methods section. We have now revised the manuscript to clearly describe how we pooled samples and then analyzed the sequence data (Line: 245-246).

- Table S3: how are medium- and high-quality bins defined?

Thank you for this comment for improving the method description. We used the medium- and high-quality criteria established by the genome standards consortium (Bowers et al., 2017, *Nature Biotechnology*). High-quality bin: >90% completeness and <5%

contamination. Medium-quality Bin: $\geq 50\%$ completeness and $< 10\%$ contamination. We added these criteria in the methods section (Line: 318-320).

Reference:

Bowers, Robert M., Nikos C. Kyrpides, Ramunas Stepanauskas, Miranda Harmon-Smith, Devin Doud, T. B. K. Reddy, Frederik Schulz, et al. 2017. "Minimum Information about a Single Amplified Genome (MISAG) and a Metagenome-Assembled Genome (MIMAG) of Bacteria and Archaea." *Nature Biotechnology* 35 (8): 725–31.

- Table S3: title should say human fecal/gut metagenome (as opposed to human metagenome)
Thank you for the comment. We changed the title of the Table S3 from "Human metagenome sequencing and assembly statistics" to "Human fecal metagenome sequencing and assembly statistics".

Reviewer #2 (Comments for the Author):

In this manuscript, the authors compared the performance of the standard shotgun 2x150 paired-end sequencing, ICLR sequencing, and nanopore long-read sequencing with a microbial mock community and ten human gut microbiome samples. They suggested that the ICLR sequencing is a promising method for high-throughput and accurate assembly of microbial genomes from gut metagenomes. The study is very interesting. If ICLR sequencing continues to be optimized and commercialized, it will promote the investigation of metagenomic analysis. I have the following comments to the manuscript.

1. It should be better if the comparison of performance among ICLR assay, standard shotgun, and nanopore long-read sequencing with a microbial mock community were performed based on the sequencing depth of short reads, instead of sub-assembled reads for ICLR assay.

Thank you for your helpful comments on the comparison analysis of assembling efficacy between each method using a mock community. We agree that the depth of the short reads used in the ICLR in our mock community was very high, but it was the depth required to obtain sub-assembled reads equivalent to short and ONT reads. This study is intended to demonstrate the potential of ICLR to overcome the technical gaps such as low contiguity and low accuracy in metagenomic assembly using short and ONT reads, respectively. Therefore, it was important to standardize the depth of the input sequence for assembly in order to evaluate the specific differences of assembly in each method. We agree on the importance of performing an analysis that standardizes the short-read depth for sub-assembly of ICLR to short and ONT reads in order to evaluate the practical sequence depth in ICLR, but we believe that this is outside the scope of this study.

2. For the study in ten human gut metagenomes, a mean of 312 Gbp of marked and 123.75 unmarked reads were generated, which was the extremely higher sequencing depth than that in the standard shotgun sequencing. I doubted whether the bin N50 and the largest contigs in the standard shotgun sequencing should be improved if the same sequencing depth was used.

We thank you for your insightful suggestion to evaluate the assembly efficacy using subsampled short reads for ICLR. This study is aimed to demonstrate the potential of ICLR to overcome the technical gaps in metagenome assembly using short and ONT reads. We think the use of reduced sequence depth of short reads for sub-assembly of ICLR is intended to evaluate the practicality of sequencing costs of ICLR and is outside the scope of the current study. To avoid confusion about the aim of this study, we revised the writing to focus on our paper's claims for evaluating the contiguity and accuracy of sequences of metagenomic assembly of ICLR compared with conventional methods and to avoid claims of higher throughput of ICLR than long reads. For this reason, we have removed the description that mentions the high throughput of ICLR (Line: 23, 36, 59, 187-188). Additionally, as the reviewers point out, the sequencing depth and cost are crucial for practical use for further study. Therefore, in the revised

manuscript, we stated the necessity of further studies for evaluating the practical sequencing depth and cost for sequencing in the discussion section (Line: 168-172).

3. The section of "library preparation and sequencing" should be divided into three parts following ICLR sequencing, shotgun sequencing, and nanopore long-read sequencing.

Thank you for your comments for improving the readability of our manuscript. We have taken your suggestion to revise the method section more easily to understand, including separation of the library preparation and sequencing into three parts: ICLR sequencing, shotgun short read sequencing, and ONT sequencing (Line: 206, 211, 216, 228, 235, 240, 248, and 289).

Re: mSystems01531-24R1 (**Illumina Complete Long Read assay yields contiguous bacterial genomes from human gut metagenomes**)

Dear Prof. Ami S Bhatt:

Your manuscript has been accepted, and I am forwarding it to the ASM production staff for publication. Your paper will first be checked to make sure all elements meet the technical requirements. ASM staff will contact you if anything needs to be revised before copyediting and production can begin. Otherwise, you will be notified when your proofs are ready to be viewed.

Sincerely,
Daniel Garrido
Editor
mSystems

Reviewer #1 (Comments for the Author):

my previous concerns were adequately addressed

Reviewer #2 (Comments for the Author):

All my concerns have been addressed. I have no further comment.